# Proton Therapy and Gliomas: A Systematic Review

Isabelle Chambrelant [1], Jordan Eber [1], Delphine Antoni [1], Hélène Burckel [2], Georges Noël [1,2,*]
and Romane Auvergne [2]

1   Paul Strauss Comprehensive Cancer Center, Institut de Cancérologie Strasbourg Europe (ICANS), Department of Radiation Oncology, Unicancer, 67200 Strasbourg, France; i.chambrelant@icans.eu (I.C.); j.eber@icans.eu (J.E.); d.antoni@icans.eu (D.A.)
2   Paul Strauss Comprehensive Cancer Center, Radiobiology Laboratory, Institut de Cancérologie Strasbourg Europe (ICANS), Strasbourg University, UNICANCER, 67000 Strasbourg, France; h.burckel@icans.eu (H.B.); r.auvergne@icans.eu (R.A.)
*   Correspondence: g.noel@icans.eu

**Abstract:** Background: Gliomas are primary cerebral tumors. Radiation therapy plays a key role in their treatment but with a risk of toxicity associated with the dose to and volume of normal tissue that is irradiated. With its precision properties allowing for the increased sparing of healthy tissue, proton therapy could be an interesting option for this pathology. Methods: Two reviewers performed a systematic review of original papers published between 2010 and July 2021 following PRISMA guidelines. We analyzed disease outcomes, toxicity outcomes, or dosimetry data in four separate groups: children/adults and individuals with low-/high-grade gliomas. Results: Among 15 studies, 11 concerned clinical and toxicity outcomes, and 4 reported dosimetry data. Proton therapy showed similar disease outcomes with greater tolerance than conventional radiation therapy, partly due to the better dosimetry plans. Conclusions: This review suggests that proton therapy is a promising technique for glioma treatment. However, studies with a high level of evidence are still needed to validate this finding.

**Keywords:** proton therapy; gliomas; brain tumors; outcomes; systematic review

## 1. Background

Gliomas are brain tumors that develop from glial cells, usually oligodendrocytes and astrocytes. Their yearly incidence is approximately 5 cases per 100,000 habitants, and they may develop at any age [1]. They are further categorized according to their grade. The World Health Organization's (WHO) consensus book identifies four grades among gliomas. Pilocytic astrocytoma (WHO Grade I) and astrocytoma (WHO Grade II) correspond to low-grade gliomas and high-grade gliomas, comprising anaplastic astrocytoma (WHO Grade III) and glioblastoma multiforme (WHO Grade IV), have the worst prognosis [2]. Advances in molecular genetics have allowed for the identification of additional prognostic and/or predictive mutations and epigenetic changes, such as isocitrate dehydrogenase (IDH) mutation, chromosome 1p/19q codeletion, and methyl-guanine methyl transferase (MGMT) gene promotor hypermethylation [1], used to refine the classification [3–5].

The primary treatment for these malignancies is surgery, and complete resection conveys suitable prognostic value. Chemotherapy and radiotherapy can be used in adjuvant treatment after surgery or as an exclusive treatment when surgery is not possible. Radiation therapy plays a key role in glioma treatment, but it may have permanent or disabling side effects, including neurocognitive impairment, neurologic deficits, neurovascular compromise, neuroendocrine deficiency, and second malignancies [1,6]. However, measuring the impact of radiation on cognitive function can be challenging because deficits are often subjective and exist before treatment.

The goal of radiation oncologists is to optimize the therapeutic ratio, increasing the dose in the target volume while protecting organs at risk (OAR). Protons are particles

with mass, charge, and a dose distribution superior to that of photons. Proton therapy (PRT) is an advanced radiation technique now used with the hope of reducing radiation-induced late effects. PRT is particularly promising because it is the most conformal form of radiotherapy available, and it allows for reductions in the low and intermediate radiation doses to surrounding normal tissue outside of the target volume [7]. This is possible because a proton beam has a unique dose-deposition pattern characterized by a reduced entrance dose and minimal to no exit dose compared to conventional photon irradiation, forming a Bragg peak. The Bragg peak can be precisely placed anywhere in the patient by modulating the proton energy, and several Bragg peaks can be shifted in depth and weighted to create a spread-out Bragg peak (SOBP) [8]. This is an important feature for tumors near critical structures in the brain, such as the hypothalamus, pituitary, cochlea, and optic pathways.

Currently, there are few PRT indications (pediatric tumors, tumors of the base of the skull, patients who could benefit from a dose increase in the target volume, or those in whom it is possible to expect a decrease in the risk of a second cancer), and they represent approximately 16% of current radiotherapy indications [8]. However, implementation of PRT in clinical practice is a real challenge due to its high cost, limited availability, and lack of level 1 evidence showing superior clinical outcomes [9].

This study is a systematic review and a summary of the relevant literature on proton therapy in children and adults with low- and high-grade gliomas. We aimed to describe the clinical efficacy, short-term and long-term toxicities, and dosimetry comparisons of PRT in these four patient groups.

## 2. Methods

This systematic review was written in accordance with the Preferred Reporting Items for Systematic Reviews and Meta-Analysis (PRISMA) guidelines [10]. A research protocol was published in the PROSPERO database (registration number: CRD42021230402). References were retrieved from two databases: MEDLINE via PubMed and ScienceDirect. The MeSH search terms are shown in Table 1. An advanced search strategy was used on each search platform using the most common synonyms. Additional papers were identified by scanning the references of relevant papers.

**Table 1.** MeSH expressions used on the two databases.

| Database | MeSH Search Expression |
| --- | --- |
| PubMed | (Protons OR proton therapy) AND (glioma) |
| ScienceDirect | Title, abstract, keywords: protons AND glioma AND brain tumors |

Eligibility criteria were prospective or retrospective studies published between 2010 and July 2021 with at least one of the following criteria: clinical outcomes, toxicity outcomes or dosimetry data, and separating the data between children/adults and low-/high-grade gliomas. The exclusion criteria were as follows: case report, review article, meta-analysis, and abstract. Studies treating a variety of brain tumors and those where patients were treated with photons and then with protons were also excluded.

The titles and abstracts of studies retrieved using the search strategy were screened independently by two reviewers to identify studies that potentially met the inclusion criteria outlined above. There was no disagreement between the two reviewers over the eligibility of the studies.

## 3. Results

Our MeSH search in PubMed and ScienceDirect returned 315 references, 11 of which were duplicates. Among the 304 remaining articles, 24 were selected from the title and abstract by the two reviewers. Eleven of them were kept after a full-text review. After checking the references of these articles, four other studies were considered eligible for our review for a total of 15 original papers. Figure 1 shows the full study selection process and

reasons for exclusion. A total of 11 papers reported clinical and toxicity outcomes (Table 2) and 4 concerned dosimetry data (Table 3).

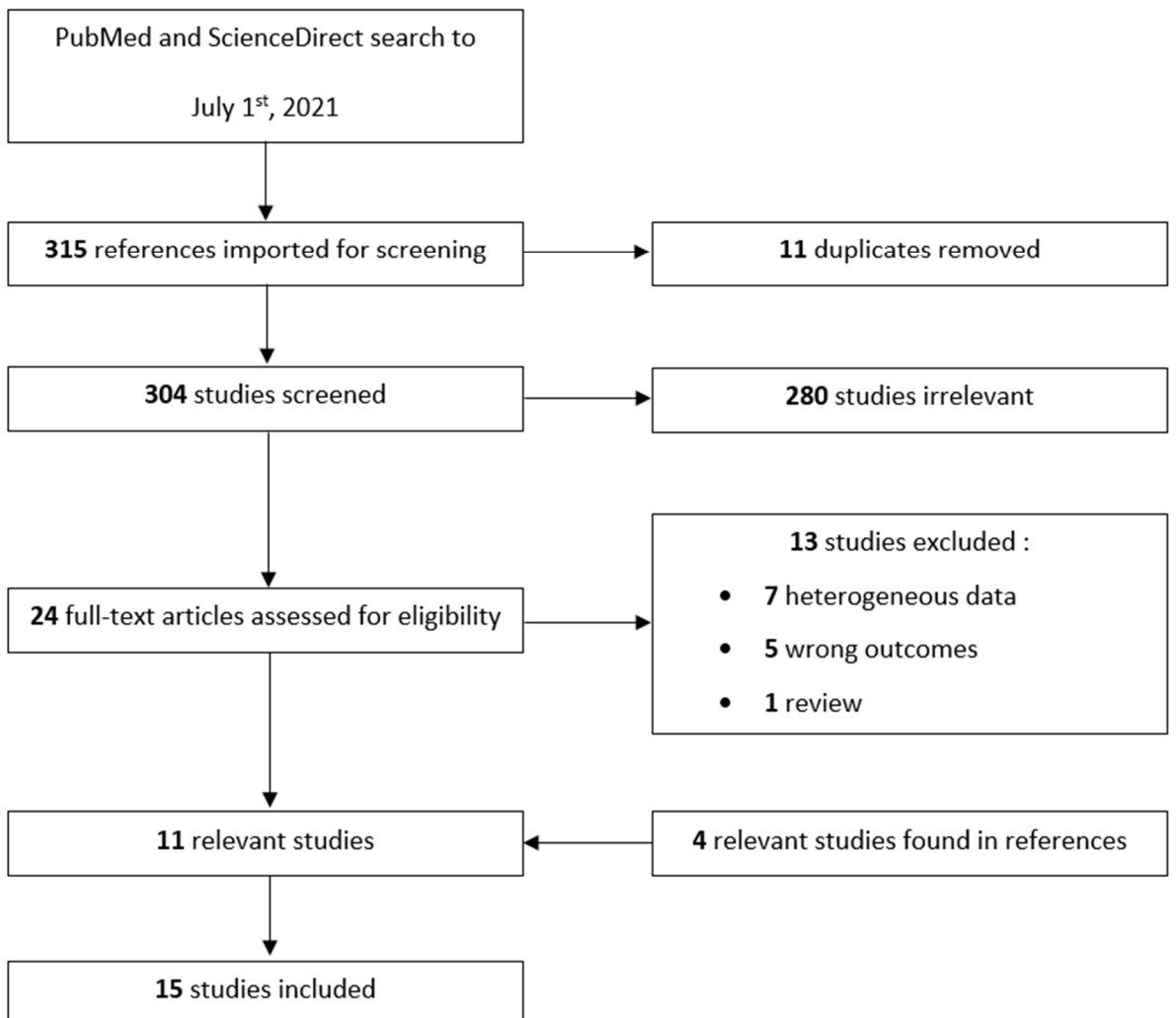

**Figure 1.** PRISMA flowchart of the literature search and study selection process.

**Table 2.** Clinical and toxicity outcomes.

| Author | Year | Type Of Study | Population | Median Age at RT (Years (Range)) | Grade | Number of Patients | Prescription | Median RT Dose (Gy RBE (Range)) | Volume | Number of Patients with Pre-RT Chemotherapy | Median Follow-up (Years (Range)) | Clinical Outcomes | Toxicity Outcomes | PsP |
|---|---|---|---|---|---|---|---|---|---|---|---|---|---|---|
| Greenberger et al. | 2014 | Retrospective | Children | 11.0 (2.7–21.5) | LGG | 32 | NA | 52.2 | CTV = GTV + 3–5 mm PTV = CTV + 8–12 mm | 16 | 7.6 (3.2–18.2) | 8-year PFS and OS rates = 83% and 100%, respectively | Significant decline in children < 7 years and those with higher dose to the left temporal lobe and hippocampus | NA |
| Mannina et al. | 2016 | Retrospective | Children | 10.9 (4–20) | LGG | 15 | NA | 54 (50.4–59.4) | NA | 9 | 4.6 | NA | NA | 3 patients (20%), the maximum volume was observed 3 to 8 months after PRT and regressed after 18 months |
| Indelicato et al. | 2019 | Prospective | Children | 9 (2–21) | LGG | 174 | 129 treated with 54 Gy RBE45 treated with <54 Gy RBE | NA | CTV = GTV + 5 mm PTV = CTV + 3 mm | 74 | 4.4 (0.5–11.4) | 5-year PFS and OS rates = 84% and 92%, respectively | 12.6% nausea or vomiting; 1.1% headaches; 2.9% sensorineural troubles; 22% neuroendocrine deficiency | 56 patients (32%) |
| Ludmir et al. | 2019 | Retrospective | Children | 10.0 (1.0–17.6) | LGG | 83 | NA | 50.4 (45–59.4) | NA | 32 | 5.6 | Improved local control for PBT patients (HR 0.34, 95% CI: 0.10–1.18, $p = 0.099$) | NA | RT modality was found to predict PsP, with a higher cumulative incidence of PsP among PBT patients (23/51, 45%) than IMRT patients (8/32, 25%) ($p = 0.048$) |
| Shih et al. | 2015 | Prospective | Adult | 37.5 (22–56) | LGG | 20 | 54 GyRBE in 30 fractions | NA | CTV = GTV + 15 mm PTV = CTV + 8 mm | NA | 5.1 (3.3–5.2) | 5-year PFS and OS rates = 40% and 84%, respectively | Patients with LGG tolerate proton therapy well, and a subset develops neuroendocrine deficiencies. There is no evidence for overall decline in cognitive function or QOL | NA |

**Table 2.** *Cont.*

| Author | Year | Type Of Study | Population | Median Age at RT (Years (Range)) | Grade | Number of Patients | Prescription | Median RT Dose (Gy RBE (Range)) | Volume | Number of Patients with Pre-RT Chemotherapy | Median Follow-up (Years (Range)) | Clinical Outcomes | Toxicity Outcomes | PsP |
|---|---|---|---|---|---|---|---|---|---|---|---|---|---|---|
| Bronk et al. | 2018 | Retrospective | Adult | 47 (24–71) Oligo 46 (26–53) Astro | LGG | 36 | NA | 54 (40–57) Oligo 50.4 (50.4–57) Astro | CTV = GTV + 10–15 mm | NA | NA | NA | NA | Same incidence of PsP in both groups (17%). The median time of PsP detection was 33 days (range, 18–116 days) |
| Tabrizi et al. | 2019 | Prospective | Adult | 37.5 (22–56) | LGG | 20 | 54 GyRBE in 30 fractions | NA | CTV = GTV + 15 mm PTV = CTV + 8 mm | NA | 6.8 (1.8–11.5) | Median PFS = 4.5 years | The majority of patients with LGG who received proton therapy retained stable cognitive and neuroendocrine function | NA |
| Dworkin et al. | 2019 | Retrospective | Adult | 37 (18–68) | LGG | 119 | NA | 54 (54–60) | NA | NA | 4.8 | NA | NA | 43.6%, the median time of PsP detection was 7.6 months (range 0.6–65.8 months). There was an increased risk of PsP following PRT + TMZ vs. PRT-alone (HR = 2.2, *p* = 0.006) |
| Muroi et al. | 2020 | Retrospective | Children | 5.8 (4–9.9) | HGG | 12 | 54 GyRBE in 30 fractions | NA | CTV = GTV + 5–10 mm PTV = CTV + 2–3 mm | NA | NA | Median PFS = 5 months (range 1–11 months), and median OS = 9 months (range 4–48 months) | The most reported toxicities were grade ≤ 2 and included alopecia in the irradiated area (n = 12), nausea(n = 4), a decreased lymphocyte count (n = 4), vomiting (n = 2), bullous dermatitis (n = 1), and allergic reaction (n = 1) | NA |

**Table 2.** *Cont.*

| Author | Year | Type Of Study | Population | Median Age at RT (Years (Range)) | Grade | Number of Patients | Prescription | Median RT Dose (Gy RBE (Range)) | Volume | Number of Patients with Pre-RT Chemotherapy | Median Follow-up (Years (Range)) | Clinical Outcomes | Toxicity Outcomes | PsP |
|---|---|---|---|---|---|---|---|---|---|---|---|---|---|---|
| Petr et al. | 2017 | Retrospective | Adult | (54.9 ±14.0 years) | HGG | 67 | 60 GyRBE in 30 fractions | NA | CTV = GTV + 20 mm PTV = CTV + 5 mm | NA | NA | NA | NA | NA |
| Brown et al. | 2021 | Prospective | Adult | 53 (26–82) IMRT 54.5 (33–72) PRT | HGG | 67 | 60 Gy or GyRBE in 30 fractions | NA | CTV = GTV + 20 mm PTV50 = CTV + 3–5 mm and PTV60 = GTV + 3–5 mm | NA | 48.7 (7.1–66.7) | Median PFS = 8.9 months in IMRT vs. 6.6 months in PRT ($p$ = 0.24), and median OS = 21.2 months in IMRT vs. 24.5 months in PRT ($p$ = 0.60) | There was no significant difference in time to cognitive failure between treatment arms. PRT was associated with a lower rate of fatigue | NA |

**Table 3.** Dosimetry data.

| Author | Year | Type of Study | Population | Median Age at RT (Years (Range)) | Grade | Number of Patients | Prescription | Median RT Dose (GyRBE (Range)) | Volume | Target Volume | Conclusion |
|---|---|---|---|---|---|---|---|---|---|---|---|
| Harrabi et al. | 2016 | In silico | Children and adult | 31.2 (2.0–64.2) | LGG | 74 | 54 GyRBE in 30 fractions | 54.0 (50.4–60) | CTV = GTV + 10 mm | Median = 185.2 cc (range 11.8–709.6) | Reduction in dose in critical neurologic structures with PRT, with similar target volume coverage in both plans |
| Eekers et al. | 2018 | In silico | Children | NA | LGG | 25 | 50.4 GyRBE in 30 fractions | NA | CTV = GTV + 10 mm<br>PTV = CTV + 2 mm | Mean = 240 cc (range 92–456) | IMPT was better than the other modalities to spare OAR, especially those located contralateral to the target volume |
| Dennis et al. | 2013 | In silico | Adult | NA | LGG | 11 | 54 GyRBE in 30 fractions | NA | CTV = GTV + 15 mm<br>PTV = CTV + 3 mm | Mean = 162.2 cc (range 22.5–390.3) | Equivalent uniform dose (EUD) between 10 and 20 GyRBE lower with PRT to crucial neuronal structures, including optic nerves, hippocampus, cochlea, and pituitary |
| Adeberg et al. | 2016 | In silico | Adult | 36.5 (26–63) | HGG | 12 | 60 GyRBE in 30 fractions | 60 (56.0–60.0) | CTV = GTV + 20–30 mm | NA | Statistically significant reductions of mean dose ($D_{mean}$) with IMPT in neurosensorial structures, neuroendocrine structures, and critical organs of neurocognition ($p < 0.05$) |

### 3.1. Low-Grade Gliomas

Low-grade gliomas (LGGs) are brain tumors with moderate malignancy grade (WHO Grade II), and their progression is slow. These are the most common brain tumors in children. They have become the third most common pediatric brain tumor type treated with PRT worldwide after medulloblastomas and ependymomas [11]. Current treatments convey a long-term survival rate that exceeds 90%. LGGs are relatively rare primary brain tumors in adults (incidence approximately 1 per 100,000 habitants). The mean age at diagnosis is 39 years [12]. Most of these patients survive with their disease for >5 years. Therefore, therapeutic strategies must also minimize late effects and prioritize preserving quality of life (QOL) [13].

#### 3.1.1. Children

##### Disease Outcomes

Indelicato et al. [14] analyzed data from 174 pediatric patients identified as having a nonmetastatic LGG with a minimum of 6 months of potential follow-up. Because treatment guidelines were modified, 129 (74%) and 45 (26%) patients received 54 Gy RBE, and 50.4 Gy RBE, respectively. They defined gross tumor volume (GTV) by the gross disease at the time of radiation. The clinical target volume (CTV) was defined by GTV + 5 mm. The planning target volume (PTV) was CTV + 3 mm. With a median follow-up of 4.4 years (range, 0.5–11.4), the 5-year progression-free survival (PFS) and overall survival (OS) rates were 84% (95% confidence interval (CI), 77–89%) and 92% (95% CI, 85–95%), respectively. On univariate analysis, young age (<6 years old) at diagnosis was associated with improved 5-year OS (97% vs. 86%, $p < 0.05$). Radiotherapy dose and tumor subsite were significantly correlated with local control (LC) and PFS. Patients who received 54 Gy RBE had a better LC rate (91% vs. 67%, $p < 0.001$) and PFS rate (90% vs. 67%, $p < 0.001$) for those who received <54 Gy RBE, with the dose effect more pronounced in children <6 years old. LC and PFS rates were greater for LGGs of the cerebellar or cerebral hemispheres or tumors along the midline of the supratentorial brain compared to those of the brainstem or spinal cord ($p < 0.01$) [14].

Greenberger et al. reported clinical outcomes for 32 patients with a median follow-up of 7.6 years (range, 3.2–18.2 years). The median age at treatment was 11 years. The median radiation therapy (RT) dose was 52.2 Gy RBE (range, 48.6–54 Gy RBE). GTV was defined as the resection cavity and any gross tumor visible on MRI or CT, including fluid-attenuated inversion recovery (FLAIR) or T2 signal change abnormality involving the optic radiation. CTV included GTV + 3–5 mm. PTV was CTV + 8–12 mm. In this study, 8-year PFS and OS were 83% and 100%, respectively [15].

##### Toxicity Outcomes

In the study of Indelicato et al., major acute toxicities included nausea or vomiting (12.6%) and headaches (1.1%). Four percent of patients were affected by serious late effects. Five patients (2.9%) experienced sensorineural issues (visual or hearing). Thirty-nine patients (22%) developed neuroendocrine deficiency (primarily growth hormone), and 6 (3.5%) experienced asymptomatic vasculopathy [14].

Neurocognitive outcomes were assessed using several scales from the study of Greenberger et al., and for the analysis, patients were stratified by age (or 7 years) and by tumor location. They did not show significant neurocognitive declines as a whole population, but a significant decline was observed in children younger than 7 years and those with higher doses to the left temporal lobe and hippocampus. Stabilization or improvement in visual acuity occurred in 83.3% of patients. A mean dose to the pituitary and hypothalamus of greater than or equal to 40 Gy RBE was correlated with neuroendocrine deficiencies. Two patients (6.2%) developed vasculopathy [15].

Pseudoprogression

Pseudoprogression (PsP) can be defined as new contrast enhancement or its enlargement within the radiation field that spontaneously resolves with modifying therapy [16].

Mannina et al. described PsP in three patients (20%) treated with PRT for juvenile pilocytic astrocytomas. The maximum volume was reached between 3 and 8 months after PRT and regressed after 18 months [17].

Indelicato et al. observed PsP in 56 patients (32%) with a mean onset of 12.1 months after PRT. Of these 56 patients, 32% were symptomatic and required treatment [14].

In a large series of 83 pediatric LGG patients, Ludmir et al. reported PsP in 32 (39%) photon-based intensity-modulated RT (IMRT) and 51 (61%) PRT-treated patients. They observed a statistically significant difference in the incidence of PsP, with 25% and 45% in the IMRT and PRT groups, respectively ($p = 0.048$). An RT dose >50.4 Gy RBE predicted higher rates of PsP ($p = 0.016$). The median time of PsP detection was 4 months (range, 1–8 months) after treatment [18].

### 3.1.2. Adults

Disease Outcomes

Shih et al. reported survival outcomes and potential treatment-associated morbidity in a prospective cohort of 20 adults (median age, 37.5 years) with LGG who received PRT. They received passive scattering PRT at a dose of 54 Gy RBE in 30 fractions. CTV was defined as the composite of the T2-hyperintense tumor, any T1-enhancing disease, and the abutting surgical bed + 15 mm. PTV was CTV + 8 mm. With a median follow-up of 5.1 years, the OS and PFS at 5 years were 84% and 40%, respectively [19].

In 2019, an update report of the Shih et al. study [19] was published, with a minimum follow-up of 5 years on the 14 patients alive. With a median follow-up of 6.8 years, the median PFS was 4.5 years [20].

Toxicity Outcomes

Major acute toxicities in the study by Shih et al. included fatigue (100%), alopecia (85%), scalp erythema (85%), and headache (75%). There was no grade 4 or 5 acute or late toxicities reported. The most common long-term toxicities were headaches (75%), fatigue (85%), and alopecia (60%), but 11 patients were not considered in this analysis because they were removed at the time of tumor progression. Eight patients exhibited baseline neurocognitive impairment (language, visual or verbal memory, processing speed). Compared to baseline values, cognitive function was stable with a median follow-up of 3.2 years. Scores for the Quality-of-Life questionnaires, Becks Depression Inventory, and Becks Anxiety Inventory remained constant, and the patients' employment was not impacted. Finally, they reported a 15%, 25%, and 30% risk for developing hormone deficiencies at 1, 3, and 5 years, respectively. The primary neuroendocrine disturbances were central hypothyroidism, adrenal insufficiency, and central hypogonadism [19].

Results of the update by Tabrizi et al. confirmed excellent PRT tolerance. They demonstrated that neuroendocrine deficiencies tended to be more common in patients who received ≥ 20 Gy RBE to the hypothalamus or pituitary ($p = 0.142$), with an inverse relationship between dose and time of onset [20].

Pseudoprogression

Bronk et al. reported PsP outcomes of 36 patients with LGG treated with IMRT (50%) and PRT (50%). GTV was defined as the surgical cavity and any residual contrast-enhancing or noncontrast-enhancing tumor. CTV was defined as GTV + 1–1.5 cm. The same incidence of PsP was observed in both groups (17%). The median time of PsP detection was 33 days (range, 18–116 days) [21].

In a larger series of 119 patients, Dworkin et al. observed PsP in 43.6% of cases. The median time of PsP detection was 7.6 months (range, 0.6–65.8 months) and PsP was more prevalent following PRT + TMZ versus PRT-alone (HR = 2.2, $p = 0.006$) [22].

### 3.1.3. Dosimetry Data

In 2013, Dennis et al. reported a dosimetry comparison of proton and photon IMRT plans in 11 adult patients with LGGs. GTV was defined as the surgical cavity, any gadolinium enhancement, and T2 hyperintense findings. CTV was defined as GTV + 1.5 cm, and PTV was defined as CTV + 3 mm. The prescription dose was 54 Gy RBE in 30 fractions, and the mean target volume was 162.2 mL (range, 22.5–390.3). They demonstrated, on average, an equivalent uniform dose (EUD) between 10 and 20 Gy RBE lower with PRT to crucial neuronal structures, including the optic nerves, hippocampus, cochlea, and pituitary. However, the difference between normal tissue complication probability (NTCP) for protons and IMRT was not significant. Moreover, PRT decreased the risk of radiation-induced second intracranial tumors (47 vs. 106 per 10,000 cases per year) [23].

In a larger cohort of 74 patients (children and adults combined) with LGGs, Harrabi et al. compared conventional three-dimensional RT (3D-CRT) and PRT plans, with a median dose of 54 Gy RBE in 30 fractions. The initial GTV was defined as a hyperintense low-grade tumor mass, surgical resection cavity, and perifocal edema on T2-FLAIR. CTV was defined as GTV + 1 cm. The median target volume was 185.2 mL (range, 11.8–709.6). They also illustrated a reduced dose in critical neurologic structures with PRT, with similar target volume coverage in both plans [24].

In 2018, the Radiation Oncology Collaborative Comparison (ROCOCO) group conducted an international multicenter in silico treatment planning study. For 25 LGG patients, they generated four plans, with a total dose of 50.4 Gy RBE: IMRT, volumetric modulated arc therapy (VMAT), tomotherapy (TOMO), and intensity-modulated proton therapy (IMPT). GTV was delineated as the resection cavity, encompassing any residual/recurrent macroscopic tumor on the planning CT fused with the (pre- and postsurgical) MRI (T1-weighted with contrast agent (CA) and T2-weighted/FLAIR images). CTV was GTV + 1 cm, and PTV was CTV + 2 mm for VMAT, IMRT and IMPT or CTV + 3 mm for TOMO. The mean target volume was 240 mL (range, 92–456), with a median of 171 mL, and they observed excellent coverage for all techniques ($V_{95\%}$ range, 99.5–100%) with a statistically significant advantage for TOMO compared to VMAT ($p = 0.02$; $V_{95\%} = 99.9\%$). They also demonstrated that IMPT was better than the other modalities for sparing OAR, especially those located contralateral to the target volume. Notably, in this study, the brain volume receiving 20 Gy RBE was statistically significantly reduced using IMPT compared to the overall plans, with $V_{20GyRBE} = 39$ mL vs. 56.1, 55, and 52.1 mL for VMAT, TOMO, and IMRT, respectively ($p < 0.02$) [25].

### 3.2. High-Grade Gliomas

High-grade gliomas (HGGs) include anaplastic astrocytoma and glioblastoma multiforme (GBM). GBM is the most common primary malignant brain tumor in adults. The standard therapy is maximal surgical resection followed by RT with concurrent and adjuvant chemotherapy using temozolomide (TMZ) [26,27]. The median survival is approximately 15 months and varies depending on molecular markers [1,28,29].

In patients with HGGs, the rationale for PRT may be different. Indeed, in these highly aggressive tumors, the primary goal is to improve patient prognosis. PRT is interesting here because, in addition to its advantages in protecting healthy tissues, it allows for an increase in the dose to the target volume.

### 3.2.1. Children

One reference was found for children with HGGs treated with PRT. Muroi et al. reported clinical and toxicity outcomes of 12 children with diffuse intrinsic pontine glioma (DIPG). They received PRT at a dose of 54 Gy RBE in 30 fractions with concurrent TMZ. CTV was defined as the area of hyperintensity on T2-weighted images + 5–10 mm. PTV was CTV + 2–3 mm. The median OS and PFS were 9 months and 5 months, respectively. Treatment was well tolerated by most patients, and most reported toxicities were grade $\leq$ 2 [30].

### 3.2.2. Adults

Petr et al. have studied brain volume and perfusion changes in healthy tissue in a prospective cohort of 67 patients with GBM who were treated with an adjuvant photon (70%) or proton (30%) radiochemotherapy in combination with TMZ. GTV contained the surgical cavity and macroscopic tumor. CTV was GTV + 20 mm and PTV was CTV + 5 mm. They showed that the brain tissue volume decrease was higher with photons than protons ($p = 3 \times 10^{-4}$) and was dependent on the radiation dose delivered (0.9% per 10 Gy, $p = 1 \times 10^{-5}$). However, the decrease in perfusion was not significantly different [31].

In a recent prospective and randomized study, Brown et al. reported disease and toxicity outcomes in 67 adult patients with glioblastoma treated either with photons or protons. They received radiations at a dose of 60 Gy or Gy RBE in 30 fractions with concurrent TMZ. CTV was defined as tumor cavity and any residual T1 tumor enhancement + 20 mm. They used a simultaneous integrated boost technique to treat both the PTV50 (CTV + 3–5 mm) and PTV60 (GTV + 3–5 mm) to 50 and 60 Gy in 30 fractions, respectively. The median OS and PFS were not significantly different in both arms. There were also no statistically significant differences in the rates of deterioration between the two treatment arms at 6 months, and PRT was not associated with a delay in time to cognitive failure. There was a higher incidence of patient-reported fatigue with photon therapy (24% vs. 58%, $p = 0.05$) [32].

### 3.2.3. Dosimetry Data

Only one study comparing three dosimetry plans (IMPT, VMAT, and 3D-CRT) in 12 HGG patients was found. The median dose was 60 Gy RBE (range, 56.0–60.0 Gy RBE) in 30 fractions. GTV was defined as the contrast-enhancing lesion visible with T1-weighted MR imaging and T2-FLAIR hyperintense areas. CTV included GTV + 2–3 cm margin representing the surgical resection cavity and perifocal edema-respecting anatomic borders. There was no PTV. While the target coverage was similar in overall modalities, Adeberg et al. showed statistically significant reductions in the mean dose ($D_{mean}$) with IMPT compared to VMAT and 3D-CRT in neurosensorial structures (i.e., contralateral optic nerve), neuroendocrine structures (i.e., pituitary gland), and critical organs of neurocognition (i.e., ipsilateral hippocampus, contralateral subventricular zone, and whole brain) ($p < 0.05$) [33].

## 4. Discussion

In this work, we highlighted some of the advantages of PRT for the treatment of gliomas. Indeed, the use of PRT for this indication shows suitable clinical outcomes with an acceptable tolerance profile and better dosimetry plans than photon therapy.

In LGG pediatric patients, two studies reported similar PFS and OS with PRT, approximately 84% and between 92% and 100%, respectively [14,15]. Values were lower in adult patients, with 40% and 84% for PFS and OS, respectively. Among the prognostic factors, tumor location in the brainstem/spinal cord and dose < 54 Gy RBE seem to be associated with poorer local control [14]. However, these results are all consistent with those of photon therapy [13,34–36]. Moreover, Jhaveri et al. showed that glioma patients treated with PRT had a superior median and 5-year survival compared to patients treated with photon therapy: 45.9 vs. 29.7 months ($p = 0.009$) and 46.1% vs. 35.5% ($p = 0.0160$), respectively. In multivariable analysis, this observation persisted for both the LGG and HGG subgroups with a lower risk of death with PRT (hazard ratio (HR) = 0.46, CI (0.22–0.98), $p = 0.043$ and HR = 0.67, CI (0.53–0.84), $p < 0.001$, respectively) [37]. Although PRT provides dosimetry precision, allowing a reduced dose to healthy tissue, the available data do not suggest an increased risk of local failure. A prospective phase II study (NCT01358058) is ongoing in the United States to assess the progression-free survival of PRT for LGG and favorable grade 3 gliomas. The study completion is planned for August 2022 [38].

PRT for LGG did not result in a significant decline in cognitive function in pediatric [14,15] or adult [20] patients. However, younger patients and patients who received a higher dose of PRT to the left temporal lobe or hippocampus appeared to be more ad-

versely affected [15]. Jalali et al. also demonstrated that these factors were predictors of neurocognitive decline with photon therapy [39]. Therefore, it may be interesting to monitor these patients with more frequent neurocognitive tests for the early detection and management of deficits. Although an updated study, the largest median follow-up remains as 6.8 years [20], and we need more hindsight to validate the long-term safety and tolerance of PRT for gliomas. One randomized phase II study by the NRG Oncology research group (NCT03180502) aims to randomize 120 LGG adult patients to receive either proton therapy or IMRT to measure changes in cognitive function. The estimated completion date of this study is January 2030 [40].

Neuroendocrine dysfunctions are well-recognized sequelae of brain photon radiotherapy in pediatric and adult patients [41,42]. We found similar data with PRT in this review. Greenberger et al. reported a limit dose of 40 Gy RBE correlated with these altered effects in children [15], while Shih et al. found a lower dose of 20 Gy RBE in adults [19]. The incidence increases with time after radiation therapy. Future dosimetry studies with larger cohorts are important to determine dose constraints to neuroendocrine structures and other OARs.

Based on its specific dose distribution, PRT offers low toxicity and reduces the second malignancy risk by a factor of 2 to 10 compared to photon therapy in dose modeling studies [43], which is supported by clinical data [44]. These data have a significant impact, particularly in LGG patients who can be long survivors.

The incidence of PsP in LGG subjects following PRT is not negligible. Both pediatric and adult subjects can present with PsP in up to one out of every three cases. Five studies in our review reported data about PsP. Ludmir et al. were the first to demonstrate a relationship between PsP incidence and RT modality (45% vs. 25% with protons and photons, respectively, $p = 0.048$) and RT dose (>50.4 Gy RBE, $p = 0.016$) in children with LGG [18]. This observation is not found by all authors [21]. Moreover, the median time of PsP detection is very disparate, between 1 and 12 months [14,17,18,21,22]. In a recent systematic review, the incidence of PsP after PRT in pediatric LGG was estimated to be 34% (95% CI, 23–45%), and there was no statistically significant difference in the PsP incidence rate between modalities (P-heterogeneity = 0.96). For adult LGG, the incidence of PsP following PRT was significantly higher than that following IMRT (P-heterogeneity = 0.04) [45]. These results may be explained by the lack of data and the lack of standardized diagnostic criteria for PsP, resulting in an unknown degree of heterogeneity [16,46].

In HGG pediatric patients, only one study was found that dealt specifically with DIPG. The authors compared their series with a historical group treated with photon therapy. There were no significant differences in PFS and OS between the groups [30]. The results were the same for HGG adult patients, with also no significant difference in time to cognitive failure between treatment arms [32]. This may be explained by the very aggressive nature of these tumors. This does not allow to take advantage of the benefits of the PRT. However, the dosimetric advantages of PT in sparing dose to OARs allow for dose escalation to potentially improve tumor control and survival outcomes. An ongoing randomized trial (NCT02179086) is assessing the potential survival benefit of dose-escalated PRT compared to standard dose photons in the treatment of GBM [47].

Most of the studies in our review are retrospective, and this is a major bias, especially for clinical data. However, some retrospective studies are useful to provide some information not available by prospective trials that remain limited in terms of number. Furthermore, information reported by retrospective studies can be an interesting background for future prospective trials.

In silico studies indicate that proton therapy has the potential to spare OARs while preserving the dose administered to the target volume [23–25,33]. However, data must be interpreted with caution because treatment plans were prepared in different institutes, with their own protocols of delineation, prescription, and treatment planning. Moreover, we noticed that the definition of target volumes was different in each trial.

## 5. Conclusions

The literature concerning PRT and gliomas is poor, and the populations are often heterogeneous. Most of the data we found was from retrospective studies with a low level of evidence. Improved knowledge of the technical abilities and limitations of both photon and proton dosimetry is necessary to better understand each of these techniques. Currently, LGG pediatric patients share all critical characteristics where the benefits of PRT can be proved: high likelihood of long-term control and survival and the significant sensitivity of the young brain to the late effects seen with the use of photon radiation therapy, but the dose-escalated PRT in GBM is an area of future research. Although PRT is very promising, prospective clinical trials with wisely defined populations and endpoints are necessary to assess the real clinical and dosimetry benefits of PRT for glioma treatment.

**Author Contributions:** I.C.: Conceptualization, Methodology, Formal analysis, Investigation, Resources, Data curation, Writing—original draft, Writing—review and editing, Visualization. J.E.: Methodology, Formal analysis, Investigation, Resources, Writing—review and editing. D.A.: Validation, Writing—review and editing. H.B.: Validation, Writing—review and editing. G.N.: Conceptualization, Methodology, Validation, Resources, Writing—review and editing, Supervision. R.A.: Validation, Resources, Writing—review and editing. All authors have read and agreed to the published version of the manuscript.

**Funding:** This research received no external funding.

**Institutional Review Board Statement:** Not applicable.

**Informed Consent Statement:** Not applicable.

**Data Availability Statement:** Not applicable.

**Conflicts of Interest:** The authors declare no conflict of interest.

## Abbreviations

The following abbreviations are used in this manuscript:

| | |
|---|---|
| 3D-CRT | Conventional three-dimensional RT |
| CTV | Clinical target volume |
| DIPG | Diffuse intrinsic pontine glioma |
| EUD | Equivalent uniform dose |
| FLAIR | Fluid-attenuated inversion recovery |
| GBM | Glioblastoma multiforme |
| Gy RBE | Gray radiobiological equivalent |
| GTV | Gross tumor volume |
| HGGs | High-grade gliomas |
| IMPT | Intensity-modulated proton therapy |
| IMRT | Intensity-modulated RT |
| IDH | Isocitrate dehydrogenase |
| LET | Linear energy transfer |
| LC | Local control |
| LGGs | Low-grade gliomas |
| MGMT | Methyl-guanine methyl transferase |
| NTCP | Normal tissue complication probability |
| OAR | Organs at risk |
| OS | Overall survival |
| PTV | Planning target volume |
| PFS | Progression-free survival |
| PRT | Proton therapy |
| PsP | Pseudoprogression |
| QoL | Quality of life |
| RT | Radiation therapy |
| RBE | Relative biological effectiveness |
| SOBP | Spread-out Bragg peak |

| TMZ | Temozolomide |
| TOMO | Tomotherapy |
| VMAT | Volumetric modulated arc therapy |
| WHO | World Health Organization |

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
