# Peer review of "Proton Therapy and Gliomas: A Systematic Review"

_radiation, doi:10.3390/radiation1030019_

Round 1

Reviewer 1 Report

Authors revised their manuscript, so my decision is "Accept in present form".

This manuscript is a resubmission of an earlier submission. The following is a list of the peer review reports and author responses from that submission.

Round 1

Reviewer 1 Report

The authors review the literatures concerning proton beam therapy for children and adults with glioma and suggest proton therapy is a promising for them. However, only 11 literatures were selected in the review, although some of unselected important studies seem to exist.

Please recheck following examples;

Pseudo-progression:

Increase of pseudoprogression and other treatment related effects in low-grade glioma patients treated with proton radiation and temozolomide. Dworkin M, et al. J Neurooncol. 2019.

Extended volumetric follow-up of juvenile pilocytic astrocytomas treated with proton beam therapy. Mannina EM. International Journal of Particle Therapy 2016

Clinical outcomes:

Proton therapy for newly diagnosed pediatric diffuse intrinsic pontine glioma. Muroi A, et al. Nerv Syst. 2020

Proton beam therapy with concurrent chemotherapy for glioblastoma multiforme: comparison of nimustine hydrochloride and temozolomide. Mizumoto M, et al. J Neurooncol. 2016 

Proton vs. Photon Radiation Therapy for Primary Gliomas: An Analysis of the National Cancer Data Base. Jhaveri J, et al. Front Oncol. 2018

Furthermore, a paper with results of an important randomized phase II study (IMRT vs. Proton) was recently published (Neuro Oncol, online ahead of print), although papers published between 2010 and 2020 were searched in this review.

They attempt to summarize pseudo-progression and clinical outcomes after proton therapy by dividing into low-grade and high-grade gliomas, but important points and summary of the study are unclear. Please make their summary and opinion clear to enhance value of the review.

Reviewer 2 Report

I think interesting review, but there are some points to revise before publishing.

  1. Protons are not heavy particle. Heavy particle therapy is , for example, carbon-ion radiotherapy. Authors should revise introduction.
  2. I agree about author’s opinion regarding uncertainty of RBE. However, I don’t understand why authors insert this paragraph, because the explanation was not enough. Authors should explain more detail or remove this paragraph.
  3. There are many biases in retrospective studies. So, I think you should use only prospective studies when you discuss about clinical data.
  4. I think there are any other articles, for example “Phase I/II trial of hyperfractionated concomitant boost proton radiotherapy for supratentorial glioblastoma multiforme”. Why authors removed this article? I would like to know why, and I’m afraid that you missed some papers.